

# Foraging dynamics are associated with social status and context in mouse social hierarchies

Won Lee[1], Eilene Yang[1] and James P. Curley[1,2,3]

[1] Department of Psychology, Columbia University, New York, NY, United States of America
[2] Center for Integrative Animal Behavior, Columbia University, New York, NY, United States of America
[3] Department of Psychology, University of Texas at Austin, Austin, TX, United States of America

## ABSTRACT

Living in social hierarchies requires individuals to adapt their behavior and physiology. We have previously shown that male mice living in groups of 12 form linear and stable hierarchies with alpha males producing the highest daily level of major urinary proteins and urine. These findings suggest that maintaining alpha status in a social group requires higher food and water intake to generate energetic resources and produce more urine. To investigate whether social status affects eating and drinking behaviors, we measured the frequency of these behaviors in each individual mouse living in a social hierarchy with non-stop video recording for 24 h following the initiation of group housing and after social ranks were stabilized. We show alpha males eat and drink most frequently among all individuals in the hierarchy and had reduced quiescence of foraging both at the start of social housing and after hierarchies were established. Subdominants displayed a similar pattern of behavior following hierarchy formation relative to subordinates. The association strength of foraging behavior was negatively associated with that of agonistic behavior corrected for gregariousness (HWIG), suggesting animals modify foraging behavior to avoid others they engaged with aggressively. Overall, this study provides evidence that animals with different social status adapt their eating and drinking behaviors according to their physiological needs and current social environment.

## INTRODUCTION

Social dominance hierarchies emerge as animals measure the competitive ability of others through social interactions and individuals learn to consistently yield towards relatively dominant individuals (*Chase, 1982*; *Drews, 1993*). One of the universal characteristics of dominance hierarchies observed across species in the wild or housed in a laboratory is that resources such as space, food and mating partners are unevenly distributed with preferential access being skewed towards more dominant individuals (*Banks et al., 1979*). Variation in feeding behavior has been observed as a function of social status. In particular, more dominant individuals typically have increased access to food resources as reported in cats (*Bonanni et al., 2007*), fish (*Sloman et al., 2000*; *Alanärä, Burns & Metcalfe, 2001*;

Corresponding author
James P. Curley, curley@utexas.edu

*Montero et al., 2009*), crayfish (*Herberholz, McCurdy & Edwards, 2007*), domestic fowl (*Banks et al., 1979*), willow tits (*Ekman & Lilliendahl, 1993*), deer mice (*Farr & Andrews, 1978*), rats (*Blanchard & Blanchard, 1989*; *Melhorn et al., 2010*), goats (*Barroso, Alados & Boza, 2000*), dairy cows (*Olofsson, 1999*), and non-human primates (*Whitten, 1983*; *Deutsch & Lee, 1991*; *Saito, 1996*; *Sterck & Steenbeek, 1997*; *Vogel, 2005*; *Robbins, 2008*), although some studies do not show this pattern (*Stricklin & Gonyou, 1981*; *Moles et al., 2006*). This monopolization of resources comes about in part because in many hierarchies dominant animals must increase food intake to meet the metabolic demands associated with acquiring and maintain dominance via asserting physical aggression or producing chemical signals (*Hogstad, 1987*; *Gosling et al., 2000*; *Hurst & Beynon, 2004*; *Biro & Stamps, 2010*; *Nelson et al., 2015*). Further, dominant animals may need to invest more in feeding due to having a lower caloric efficiency (*Moles et al., 2006*) and a higher oxygen consumption rate (*Hogstad, 1987*). Conversely, subordinate animals may also experience shifts in feeding and metabolism due to experiencing social stress. For instance, chronically socially defeated mice have been found to both increase (*Bhatnagar et al., 2006*; *Foster et al., 2006*; *Chuang et al., 2011*) and decrease (*Meerlo et al., 1996*; *Becker et al., 2007*) food intake when experiencing chronic social stress.

Competition for water occurs in some species when water is an in-demand resource (*Christian, 1980*; *Razgour, Korine & Saltz, 2011*), but may also be a key feature of living in a social hierarchy even when water resources are not scarce. Indeed, dominant male rats living in social groups have a significantly higher frequency of drinking water even when it was given *ad libitum* (*Blanchard & Blanchard, 1989*). Further, in species where dominants scent-mark to attract females or mark their territories, dominant individuals may be physiologically required to intake more water than subordinates. Notably, dominant rats and mice scent-mark more frequently than subordinates (*Hurst & Beynon, 2004*) and typically have empty bladders (*Desjardins, Maruniak & Bronson, 1973*). Moreover, (*Desjardins, Maruniak & Bronson, 1973*) demonstrated that dominant mice flush intravenously injected radioactive molecules via urination significantly faster than subordinates. Conversely, subordinate rats and mice inhibit scent-marking behavior and show a decrease in daily urination volume (*Desjardins, Maruniak & Bronson, 1973*; *Drickamer, 1995*; *Wood et al., 2010*; *Nelson et al., 2015*; *Hou et al., 2016*; *Lee, Khan & Curley, 2017*). Taken together, these findings suggest that dominant rats and mice must increase their water intake relative to subordinate individuals.

Living in a social hierarchy also forces animals to adapt their behavioral patterns to maximize their fitness by paying attention to more dominant individuals. In primates, fish and mice, subordinate animals may monitor dominant animals or animals with whom they have had frequent aggressive interactions and inhibit their own social behavior accordingly (*Deaner, Khera & Platt, 2005*; *Pannozzo et al., 2007*; *Desjardins, Hofmann & Fernald, 2012*; *Curley, 2016b*). Since eating and drinking are essential activities for survival regardless of social status in groups, subordinate animals cannot completely inhibit their foraging behavior but rather may need to adjust these behaviors to avoid conflict with dominants. For example, desert baboon dominant males have stronger co-feeding relationships with other dominants than with subordinates males (*King, Clark & Cowlishaw, 2011*). In semi-free

ranging Mandrills, individuals tend to visit feeding zones at the same time more often when they are distant in the social hierarchy rather than close in ranks (*Naud et al., 2016*). In brown trout, subordinates temporally segregate their feeding time to avoid conflict with dominant males by choosing to visit food sources during less desirable times of the day (*Alanärä, Burns & Metcalfe, 2001*). Clearly dominant and subordinate animals adjust the timing of their feeding and drinking dependent upon their relative relationship to other individuals in their groups.

Previously, we have demonstrated that groups of 12 male outbred CD-1 male mice living in a complex housing system rapidly form stable and linear social hierarchies (*So et al., 2015*; *Williamson, Lee & Curley, 2016*; *Curley, 2016b*; *Williamson, Romeo & Curley, 2017*; *Lee, Khan & Curley, 2017*). Each male maintains a unique social rank and adjusts their social behavior flexibly and appropriately according to social context (*Curley, 2016b*; *Williamson, Romeo & Curley, 2017*; *Williamson et al., 2018*). Further, we recently reported that more dominant males produce and excrete higher levels of major urinary proteins (MUPs) and a higher volume of urine than subordinates (*Lee, Khan & Curley, 2017*). These findings pose several questions regarding the feeding and drinking patterns of differently ranked mice that face different physiological needs. In the current study, we addressed three specific questions regarding the foraging behavior of mice living in a social hierarchy: (i) do more dominant males eat or drink more frequently than subordinate males to account for their increased energetic demands? (ii) Do individuals choose feeding sites away from alpha males to avoid conflict? (iii) Do individuals adjust their eating and drinking times to avoid encountering more dominant males or males from whom they have received frequent aggression?

## METHODS
### Subjects and housing
#### Animals
In this study, we observed agonistic, eating and drinking behaviors of a total of 156 male outbred CD-1 mice aged 9–12 weeks. Seven-week old mice were obtained from Charles River Laboratories (Wilmington, MA, USA) and housed in groups of three for 2 weeks in standard sized cages with ad libitum standard chow and water. All mice were individually marked by dying their fur with nontoxic animal markers (Stoelting Co., Wood Dale, IL, USA). The 156 mice were assigned into 13 distinct groups of 12 males. These social groups were part of several different ongoing studies in our laboratory with the aim of analyzing blood and brain tissue post-mortem. The purpose of this study was to assess how feeding and drinking behavior varies with social status across hierarchy formation. Therefore, we acquired 24-hour observations of feeding and drinking behavior for each of the social groups (described below). Animals were undisturbed throughout the experiment.

#### Housing
On the day of group-housing 12 mice were weighed and placed into custom-built vivaria (Fig. S1; 150 × 80 cm and 220 cm high; Mid-Atlantic, Hagerstown, MD, USA). The vivarium was constructed as previously described (*So et al., 2015*; *Williamson, Lee & Curley, 2016*),

and consisted of an upper level with multiple shelves covered in pine bedding (36,000 cm$^2$ = 3 floor × 150 cm × 80 cm) and a lower level with five nest boxes filled with pine bedding (2,295 cm$^2$ = 5 cages × 27 cm × 17 cm) connected by tubes. The total surface of a vivarium is approximately 62,295 cm$^2$, providing 5,191 cm$^2$ per mouse. Standard chow and water were provided ad libitum from two locations on the top shelf of the vivarium. All animals either had no previous experience with any other animal in the group or had been previously housed with only one other male in the social group. Subjects were housed with constant temperature (22–23 °C) and humidity (30–50%) and a 12/12 light/dark cycle with white light (light cycle) on at 2,400 h and red lights (dark cycle) on at 1,200 h. We observed if any animal exhibited a sign of pain or injury every day. All procedures were conducted with approval from the Columbia University Institutional Animal Care and Use Committee (IACUC protocols: AC-AAAP5405, AC-AAAG0054).

## Data collection: agonistic behavior, eating, and drinking
### Agonistic behavior data collection
Animals were housed in groups for up to 27 days (range 10–28 days, mean = 18.31 days). An average of 1.70 h of daily behavior observations were undertaken on each group to determine the social hierarchy. Observations always occurred in the dark phase of the light cycle. Trained observers recorded all occurrences of fighting, chasing, mounting, subordinate posture and induced-flee behaviors and the identity of the dominant and subordinate individuals in each interaction (for ethogram, see Table S1). Data were collected using handheld android devices and directly uploaded to a timestamped Google Drive.

### Video data collection
On 16 unique days we mounted two GoPro cameras directly in front of the food and water hoppers on the left and right sides of each vivarium and continuously recorded eating and drinking for 24 h. We collected data on first day of group housing (Day 1) from five cohorts, recording from the time that animals were put into the vivarium. Eating and drinking in stable social hierarchies (Stable) were recorded from 11 separate cohorts between days 6 and 22. By sampling across a range of days we were able to assess if the time since group formation also affected feeding and drinking behavior. A total of three cohorts were videoed for feeding and drinking behavior on both Day 1 and post Day 6 (Stable). We controlled for this using cohort-ID as a random effect in all models where appropriate. We have previously demonstrated that all hierarchies become stable from Day 4 or 5 onwards (*Williamson, Lee & Curley, 2016*). We further confirmed stability of each social hierarchy through observation of stabilization of Glicko ratings (Fig. S2) after Day 5 of group housing as well as at least three days before and after the day of foraging behavior video recording. During to 1,440-min observation/video window, we coded the identities of those animals that drank or ate at the particular hopper (left/right side of a vivarium) during each minute bin. These data provided a measure of the number of minutes engaged in and the circadian rhythmicity of feeding and drinking behavior for each animal across a single dark/light cycle of a 24-hour period. However, as it is possible that individuals may spend different amount of time eating or drinking per visit dependent on their rank or

the day we further selected 603 eating and drinking visits (1.2% of eating bouts, 2.1% of drinking bouts, 3.3% of total visits) across all groups for duration analysis. We sampled data probabilistically with the representation of each animal in the duration dataset being weighted according to their frequency of eating and drinking.

### Inter-rater reliability

Each video was coded by two to three observers from a pool of 11 trained coders. Observers showed a high degree of inter-rater reliability (unweighted Cohen's kappa = 0.805, $p < 0.001$) (*Jacob, 1960*; *Lombard, Snyder-Duch & Bracken, 2002*; *Gamer, Lemon & Singh, 2012*).

## Statistical analysis

All statistical analyses were undertaken in R v. 3.4.3 (*R Core Team, 2017*). The statistical analysis for agonistic behavior, eating and drinking frequency and duration, and social network analysis of foraging and agonistic behaviors are described.

### Analysis of agonistic behavior data

With agonistic interaction data, we tested the linearity and stability of each social hierarchy by calculating Landau's $h$-value and triangle transitivity (*ttri*), and directional consistency (DC) and associated $p$-values derived from 10,000 Monte-Carlo randomizations (*De Vries, 1995*; *McDonald & Shizuka, 2013*) using the compete R package (*Curley, 2016a*) as well as by verifying the stabilization of individual Glicko ratings. Values and associated significance tests were determined for observational data up to the end of each day and over all observations. Individual ranks were determined through calculation of Glicko Ratings using the R package PlayerRatings (*Stephenson & Sonas, 2012*). In the Glicko Rating system (*Glickman, 1999*; *Williamson, Lee & Curley, 2016*), animals are initially assigned with 2,200 points then gain or lose points based on the number of wins and losses relative to the difference in ratings between themselves and their opponent (see *Williamson, Franks & Curley (2016)* for a more detailed description of the calculations). After each contest in each group the Glicko ratings of all animals in that group was continuously updated. Based on our behavioral observation of social hierarchy dynamics, we further categorized individuals into three social status groups using Glicko ratings. An alpha male holds the highest Glicko rating (social rank 1) in the hierarchy. Males in the subdominant social group are those with Glicko ratings higher than or equal to initial points, 2,200 but not the highest rating. The remainder of the males in the hierarchy that hold Glicko ratings lower than 2,200 are in the subordinate social group. The despotism of each alpha male was calculated using the compete R package (*Curley, 2016a*) by determining the proportion of all wins by alpha to all agonistic interaction (see *Williamson, Lee & Curley, 2016* for details) that occurred in each group up to the day of video recording. Associations between body weight measured on Day 1 of group housing and social rank were tested for each hierarchy using Spearman Rank correlation tests.

### Analysis of frequency and bout duration data of eating and drinking

We analyzed the data on frequency and duration of eating and drinking with generalized linear mixed effects models with a Bayesian Markov chain Monte Carlo (MCMC) sampling

using the MCMCglmm R package (*Hadfield, 2010*). MCMC is a reliable method to fit data with non-Gaussian distribution and is particularly powerful for accounting for overdispersion of Poisson distributed data (*Hadfield, 2010*). As our eating and drinking frequency data follows Poisson distribution with overdispersion, we adopted MCMC method for statistical analysis. We specified a Poisson family for the dependent variables of eating and drinking frequency of each individual (count data) and Gaussian family for duration of eating and drinking (continuous data). A default uninformative inverse gamma prior in the MCMCglmm library was used. We fitted all models with cohort ID as random slopes and intercepts in each model. We tested the effect of following fixed factors on eating and drinking frequency in each model with 1,000,000 iterations, 5,000 burn-in, and a thinning interval of 100: (i) the individual Glicko rank and despotism of each group on Day 1 and up to the day of eating/drinking video recording, (ii) individuals' social status group (alpha, subdominant, subordinate), (iii) dark/light phase and whether the hierarchies have been established (Day 1 vs. Stable) and an interaction between them. We then tested whether social status group as a fixed factor had effects on the following dependent variables in each model with 1,000,000 iterations, 5,000 burn-in, and a thinning interval of 20: (i) the percentage of visits made by individuals in the light phase to total visits (visits in light cycle/total visits in dark/light cycles*100), (ii) the maximum period of quiescence/inactivity of eating and drinking. With eating and drinking bout duration data, we tested if the bout duration of eating and drinking were associated with the following fixed factors with 10,000,000 interactions, 100,000 burn-in, and a thinning interval of 50: (i) the individual Glicko social rank on the day, (ii) despotism of each group, and (iii) whether the hierarchies were stabilized. In all models, we confirmed that convergence of the chains was attained by visually inspecting the MCMCglmm object plots, setting thinning intervals so that autocorrelation between samples were less than 0.10, and using a Gelman–Rubin test in the coda R package (*Plummer et al., 2016*). We tested the interactions among fixed effects and only included the interaction effects if the model with interaction terms yielded the lowest deviance information criterion (DIC). All interaction terms among fixed variables were tested and only selected when the model with interaction terms had significantly lowest DIC values. A two-tailed exact binomial test was used to test whether each mouse showed a location preference (right versus left) between the two food/water hoppers.

### Association patterns in foraging behavior and aggressive behavior

Within stable (post Day 6) cohorts we measured the association strength of foraging behavior by calculating the simple ratio association index (SRI) for each of 726 dyads (total number of dyadic relationships in the 11 stable cohorts of 12 individuals in each group) (*Whitehead, 2008*). Briefly, we placed two separate food and water dispensers on the opposite sides of vivaria. For a dyad with mouse A and mouse B, their simple ratio association index is calculated by $SRI = \frac{x}{x+yAB+yA+yB}$ where $x =$ total number of minute bins where A and B were foraging (eating or drinking) at the identical dispenser location, $yA =$ total number of minute bins with only $A$ identified, $yB =$ total number of minute bins with only $B$ identified, $yAB =$ total number of minute bins where A and

B are identified from different locations. As mice differ in their tendency to associate aggressively (gregariousness) across social ranks, we calculated HWIG (half-weight index corrected for individual gregariousness) as described in (*Godde et al., 2013*) as a measure of the association strength of agonistic behavior. As the SRI of foraging behavior followed a zero-inflated beta distribution, we used the brms package (*Bürkner, 2018*) to fit models accordingly. We tested whether the SRI of foraging is affected by types of relationship (alpha-other, other-other) and the association strength of agonistic interactions.

## RESULTS

### Social hierarchy characteristics

All cohorts formed a linear hierarchy over the housing period ($h'$ mean $= 0.78$, interquartile range (IQR) $= [0.67–0.81]$, all $p < 0.001$; *ttri* $= 0.87$ $[0.83–0.93]$, all $p < 0.001$; DC $= 0.86$ $[0.84–0.92]$, all $p < 0.001$). All 11 cohorts videoed after Day 6 had formed a stable linear hierarchy by the day of video recording ($h' = 0.71$ $[0.64 –0.80]$, all $p < 0.05$; ttri $0.91$ $[0.88 - 0.97]$ all $p < 0.05$; DC $= 0.91$ $[0.88 –0.93]$ all $p < 0.001$, stabilized individual Glicko ratings shown in Fig. S2). We were able to identify the final rank of all animals in each hierarchy using the Glicko ratings method as well as identify individual ranks on the day of eating/drinking video recording. For the 11 cohorts videoed on day 6 or later, individual ranks on the day of video recording correlated highly with final rank at the end of group housing indicating high stability (rhos $= 0.92$ $[0.88–0.95]$, all $p < 0.01$). In 9/11 groups the alpha male on the day of video recording was the same alpha male at the end of group housing. Although in the other two groups the final alpha male was ranked 2 and 3 respectively on the days of video recording, these two alpha males maintained their alpha status for at least 13 days after the day of video recording (Cohort A and E in Fig. S2). In the other two groups the final alpha male was ranked 2 and 3 respectively on the days of video recording. In the five groups recorded on day 1, 3/5 males remained the alpha male throughout the study and the other two males became second and third ranked in their respective groups (Fig. S2). The degree of alpha male despotism during the whole housing period varied across groups (0.56 $[0.38–0.63]$). For social groups videoed after Day 6, alpha-male despotism ranged between 0.36–0.85 with a median and IQR of 0.50 $[0.42–0.61]$ on the day of feeding/drinking video recording. Across all cohorts, initial body weight did not predict social rank (Spearman's rank correlation tests, all cohorts $p > 0.68$).

### Associations between individual social rank, group despotism and foraging frequency

For those groups in which we observed eating and drinking frequency on Day 1 of group housing, eating frequency did not have linear relationship with social rank (determined by individual Glicko ratings based on agonistic behavioral data collected on Day 1) (fixed effect mean $= -0.022$, 95% Bayesian credibility interval (BCI) $= [-0.047, 0.006]$, pMCMC $=0.099$, Fig. 1A) or despotism ($0.037$ $[-0.015, 0.087]$, pMCMC $= 0.099$). Mice with more dominant social status drank significantly more frequently ($-0.049$ $[-0.084, -0.018]$, pMCMC $= 0.001$, Fig. 1B) and individuals from groups with higher despotism drank water more frequently ($0.032$ $[0.007, 0.054]$, pMCMC $= 0.012$). In established stable social

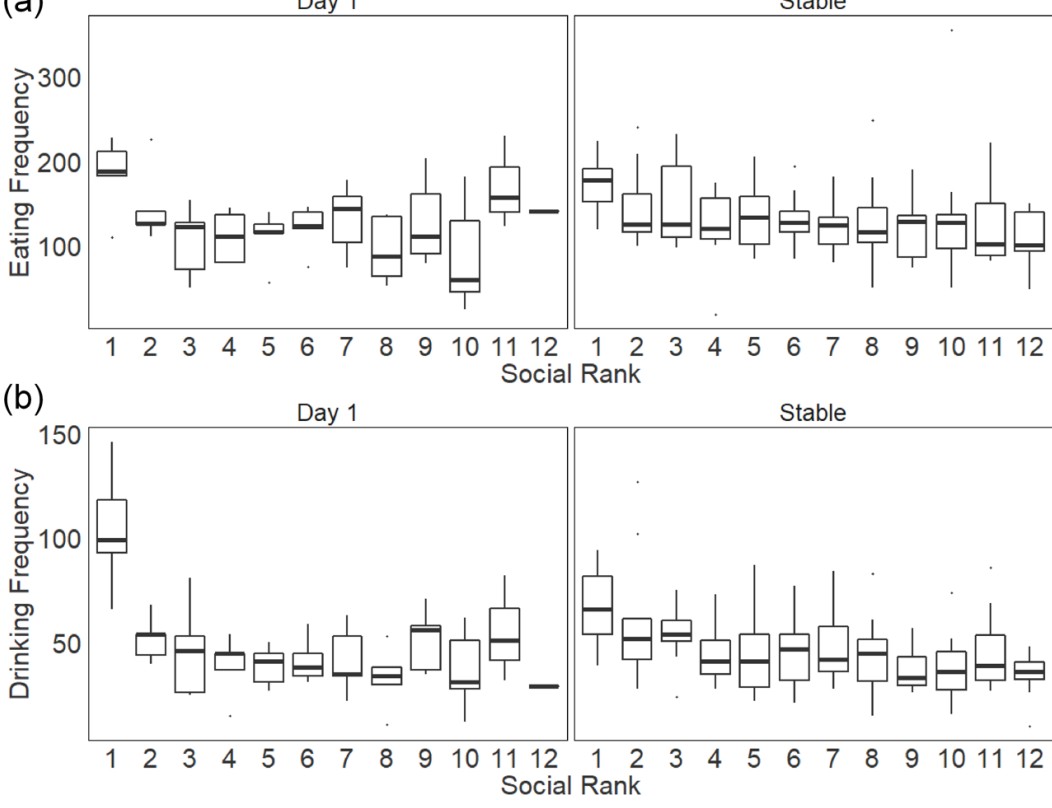

**Figure 1** **Frequency of bouts of (A) eating and (B) drinking on the day of video recording across social ranks on the first day of group housing (Day 1) and after social hierarchies were established (Stable).** Social ranks were determined by individual Glicko ratings based on agonistic behavioral data collected until the day of the video recording. Boxplots show median (horizontal bars), interquartile ranges (boxes) and 95% confidence intervals (whiskers).

hierarchies, more dominant individuals ate and drank more frequently than subordinate animals did (eating: $-0.031$ [$-0.046, -0.016$], pMCMC $< 0.001$; drinking: $-0.045$ [$-0.061$, $-0.028$], pMCMC $< 0.001$, Fig. 1). Despotism did not have an effect either on eating and drinking frequency in stable groups (eating: pMCMC $= 0.521$, drinking: pMCMC $= 0.433$). This effect was consistent across all days sampled in stable hierarchies from Day 6 to Day 22 as there was no significant effect of day on eating and drinking frequencies post Day 6 (eating: pMCMC $= 0.720$; drinking: pMCMC $= 0.498$). Mice also did not differ in eating and drinking frequency between Day 1 and the days after the hierarchies were stabilized (eating: pMCMC $= 0.381$; drinking: pMCMC $= 0.276$).

We further examined differences in eating and drinking frequency among three social status groups: alpha (Glicko rank 1, the highest Glicko rating), subdominant (other males with Glicko ratings higher than their initial starting point) and subordinate (males with Glicko ratings less than their initial starting point) (see Table S2). On Day 1, alpha males ate and drank significantly more frequently compared to subdominant (eating: $-0.444$ [$-0.808, -0.090$], pMCMC $= 0.015$; drinking: $-0.749$ [$-1.151, -0.356$], pMCMC $< 0.001$;

Fig. S3A) and subordinate groups (eating: −0.483 [−0.788, −0.150], pMCMC = 0.004; drinking: −0.984 [−1.318, −0.606], pMCMC<0.001). Subdominant males did not differ in both eating and drinking frequency from subordinate males (eating: pMCMC = 0.741; drinking: −0.235 [−0.498, 0.0316], pMCMC = 0.083). Once hierarchies were established, alpha males still showed higher frequency of eating and drinking than subdominant (eating: −0.244 [−0.456, −0.047], pMCMC = 0.021; drinking: −0.252 [−0.467, −0.029], pMCMC = 0.024; Fig. S3B) and subordinate males (eating: −0.350 [−0.541, −0.170], pMCMC<0.001; drinking: −0.494 [−0.696, −0.298], pMCMC<0.001), but the effect sizes were diminished compared to Day 1. Notably, subdominant males drank significantly more frequently than subordinate males did (−0.243 [−0.373, −0.108], pMCMC<0.001) but did not eat more frequently than subordinate males did (−0.106 [−0.231, 0.018], pMCMC =0.100).

There were no significant differences in average eating or drinking bout duration across all ranks (eating: pMCMC = 0.106; drinking: pMCMC = 0.913; Fig. S4), but the bout duration of eating and drinking was shorter on Day 1 compared to after the hierarchies stabilized (eating: 11.943 s [3.486–20.856], pMCMC = 0.009; drinking: 2.922 s [0.730–4.986], pMCMC = 0.005. The average bout duration of eating across all ranks was 16.8s [6.4s–28.8 s] on Day 1 and 25.1 s [17.3 s–50.9 s] after Day 5. The average bout duration of drinking across all ranks was 4.7 s [3.2 s–6.4 s] on Day 1 and 7.4 s [4.5 s–9.9 s] after Day 5). This finding suggests that the observed increase in the frequency of eating and drinking in alpha males translates to significant increases in total food and water consumed compared to subdominant and subordinate males.

## Variation in eating and drinking frequency during dark and light phases

Mice were housed under a 12:12 h dark/light cycle and eating and drinking behavioral data were collected beginning with the onset of the dark cycle for 24 h. Animals of all ranks ate and drank significantly more frequently during the dark phase compared to the light phase both on Day 1 and after the hierarchies achieved stability (eating: −1.033 [−1.120, −0.870], pMCMC<0.001; drinking: −1.047 [−1.244, −0.852], pMCMC<0.001; Fig. 2). There was however significant interaction effects of dark/light phase and the stability of hierarchies (Day 1 vs. Stable) in both eating and drinking frequency (eating: 0.399 [0.199, 0.586], pMCMC<0.001; drinking: 0.400 [0.173, 0.634], pMCMC<0.001; Fig. 2). Mice ate and drank more frequently during the light phase once hierarchies stabilized compared to Day 1.

At the individual level, only 5% of mice on Day 1 (0% of alpha males, 8% of sub-dominant mice, 5% of subordinate mice) and 14% of mice in established hierarchies (0% of alpha males, 19% of sub-dominants, 13% of subordinates) ate and drank more during the light phase compared to the dark phase (Fig. S5A). Subdominant males had a higher proportion of foraging bouts in the light phase compared to subordinates on Day 1 (−8.13 [−16.09, −0.18], pMCMC =0.045) and after the hierarchies were stabilized (subordinate: −5.90 [−10.90, −1.01], pMCMC =0.020). Alpha males did not differ significantly from either subdominants or subordinates (all pMCMC>0.062). When examining the relative

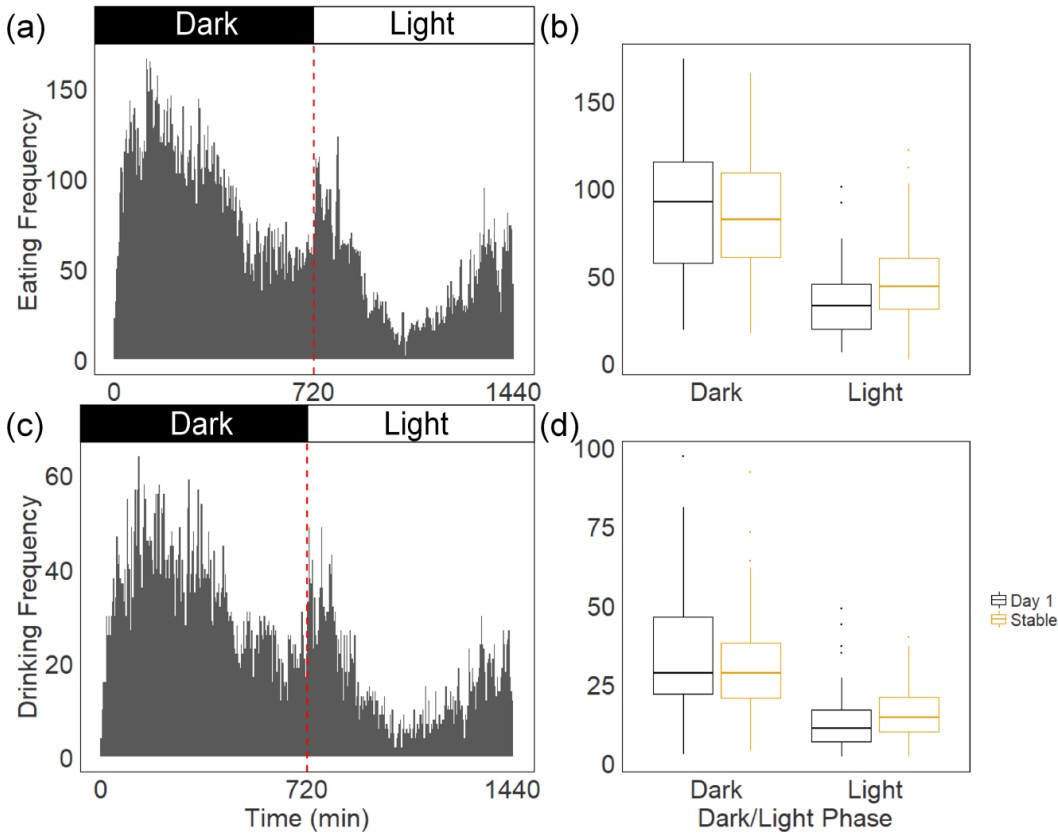

**Figure 2** **Total frequency of eating and drinking in 24 h of all individuals observed on Day 1 of hierarchy formation and after stable hierarchies were established.** (A, C) The first half of observation period was in dark cycle (minute 0 to 720) and the rest half was in light cycle (721 to 1,440). Mice eat and drink more frequently in the dark cycle both on Day 1 and after the hierarchies are established. (B, D) There was a significant interaction effect between dark/light phases and stability of hierarchies on eating and drinking frequency. Boxplots show median (horizontal bars), IQR (boxes) and 95% CI (whiskers).

frequency of eating and drinking bouts over 24 h (Fig. S5B), it is clear that subordinates show the most pronounced morning peak of foraging and alpha males are more likely to eat consistently evenly throughout the dark phase. Further, these differences are most pronounced on Day 1 of hierarchy formation compared to after hierarchy stabilization.

We also analyzed the longest duration of inactivity in eating/drinking behavior for each mouse (Fig. 3). For 82% (158 out of 192 mice) of all individuals, the longest quiescent period occurred during the light phase. 16% (30 mice) had their longest quiescent period across dark and light phases. Only 2% of all mice (4 mice) had their longest inactive period during the dark phase. On both Day 1 and after hierarchies were established, alpha males had significantly shorter quiescent periods in eating/drinking than both subdominant (Day 1: 159.8 [35.2, 285.5], pMCMC = 0.013; Stable: 79.4 [13.5, 146.8], pMCMC = 0.020) and subordinate mice (Day 1: 167.9 [55.0, 277.6], pMCMC = 0.004; Stable: 141.9 [81.6, 203.5], pMCMC<0.001). On Day 1, subdominants did not differ from subordinates in the duration of the longest quiescent periods (8.5 [−68.4, 86.2], pMCMC = 0.823) while
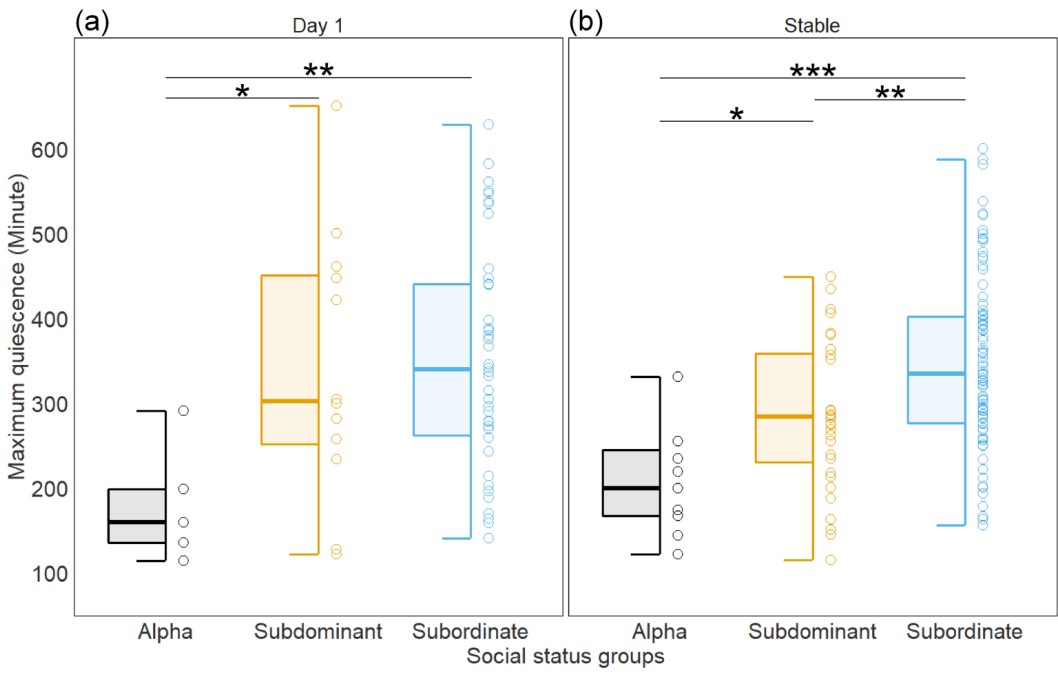

**Figure 3** **The effect of social status on maximum length of inactivity (quiescence) in eating and drinking by social status group (A) on Day 1 and (B) in stabilized hierarchies.** Boxplots show median (horizontal bars), IQR (boxes) and 95% CI (whiskers). Raw data points of each group are also shown on the right side of each box plot. Significant differences between groups are shown; *$pMCMC < 0.05$, ** $pMCMC < 0.01$, ***$pMCMC < 0.001$.

having a significantly shorter quiescent period than subordinates after the hierarchies were established (62.3 [22.6, 101.1], $pMCMC = 0.002$).

## Location preference and patterns of social association while foraging

In each vivarium, mice could eat and drink from one of two hoppers. One was placed in the top right of the vivarium and the other in the top left. We used a binomial test to see if mice showed a location preference between the two dispensers. Out of 60 mice observed on Day 1, 45 mice showed significant preferences for one particular food/water location; seven animals preferred the left food hopper and 38 animals preferred the right one. Among the five cohorts observed on Day 1, the alpha males from three cohorts showed a significant location preference. Eighteen of the 27 non-alpha males in those three cohorts preferred the same location that the alphas preferred, and nine males chose to visit the other location more often. For the 11 cohorts we observed after the social hierarchies were stabilized, 96 out of 132 mice showed a location preference (left: 36, right: 60 mice). The alpha males of seven stable cohorts significantly preferred one specific location, and 42 out of 60 non-alpha males in those seven cohorts preferred the same location as their respective alpha males did (Fig. S6). It is clear that animals do not grossly avoid the alpha males simply by preferring food/water locations that are non-preferred by the alphas.

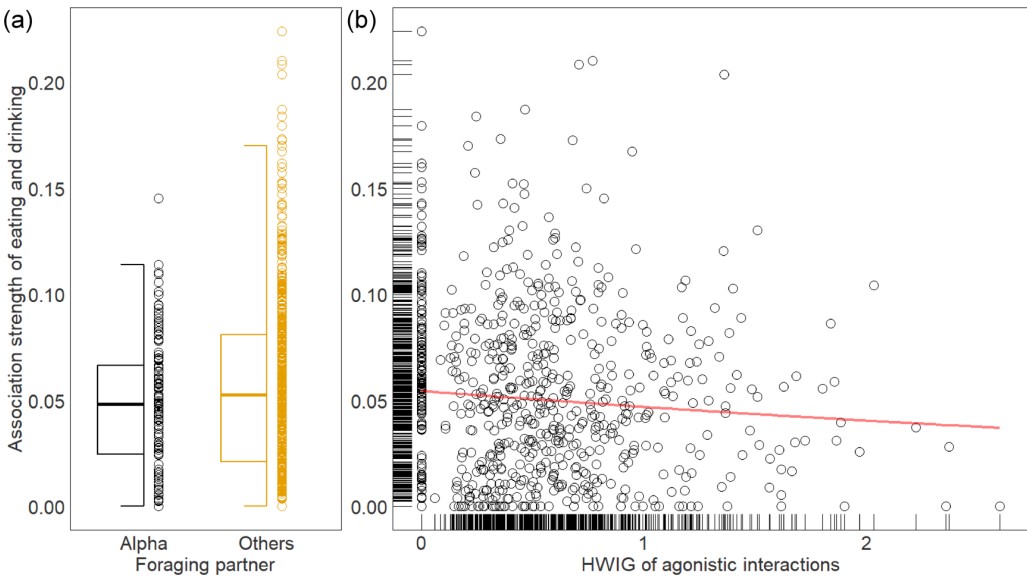

**Figure 4** **Association indices of dyadic relationships in eating and drinking behavior.** (A) Foraging association indices are not different between those of alpha-other and other-other relationships. Boxplots show median (horizontal bars), IQR (boxes) and 95% CI (whiskers). Raw data points are also shown on the right side of each box plot. (B) The association strength of foraging behavior is negatively associated with the half-weight index corrected for individual gregariousness (HWIG) of agonistic interactions. The red line indicates the fitted trend line.

Whether animals avoid associating with the alpha male in their groups while eating or drinking was more completely addressed by comparing the mean difference of association indices of eating and drinking behaviors at feeding/drinking locations between alpha-other and other-other relationships. Overall, associations were very low and individuals associated with the alpha male at a similar rate as they did with other males (0.07 [−0.01, 0.25]; Fig. 4A). Next, we tested whether the association strength of foraging behavior is related to the association strength of agonistic interactions. Since individuals vary in their tendency to associate in agonistic interactions with others, we used HWIG, a measure of the association strength of a dyadic relationship corrected for the gregariousness of both individuals (*Godde et al., 2013*). There was a significant moderate relationship between the association indices of foraging and the HWIG of agonistic interaction (−0.16 [−0.27, −0.04]; Fig. 4B), suggesting that while eating or drinking animals avoid others that they had associated frequently with in aggressive interactions.

## DISCUSSION

In this study we demonstrate that alpha male mice in social hierarchies eat and drink more than animals of all other ranks. Once hierarchies are stable, subdominant males also drink more frequently than subordinate males. Animals of all ranks visit the food and water dispensers more often during the dark phase than the light phase, though once the hierarchy is stabilized, individuals increase the proportion of eating and drinking that occurs during the light phase. Subordinate animals show the most pronounced temporal

patterning of feeding and drinking behavior with longer periods of inactivity of foraging behavior than dominant males and sharper peaks in relative activity at the onset of the dark phase. Alpha males tend to eat and drink consistently throughout the day. By analyzing the association strength of foraging behavior and agonistic behavior, we also show that animals avoid eating or drinking with others that they have exchanged aggressive interactions with rather than avoiding alpha males specifically. These findings extend our previous findings where we observed alpha males living in social hierarchies produce significantly more MUPs and urine daily suggesting that the increased food and water intake is required to meet these metabolic demands (*Lee, Khan & Curley, 2017*). Overall, this study supports the hypothesis that individuals living in a large group adapt their eating and drinking behaviors in response to physiological needs and concurrent social dynamics.

Alpha males ate more frequently than other animals on the day of hierarchy formation (Day 1) and on all days measured after hierarchies were established. Although we were not able to directly measure the amount of food and water each mouse consumed, we show that the durations of individual eating and drinking bouts across light phases is not different across social ranks, suggesting that the frequency of eating and drinking is a reliable measure of the amount of food and water each animal consumed. Dominant animals in a social hierarchy, especially the alpha male of a group, initiate and engage in a significantly higher number of aggressive interactions than relatively subordinate individuals (*Sapolsky, 1993*; *Maruska & Fernald, 2010*; *Williamson, Franks & Curley, 2016*; *Williamson, Lee & Curley, 2016*) requiring high amounts of metabolic energy (*Haller, 1995*). Moreover, (*Moles et al., 2006*) found that even when dominant and subordinate male mice do not engage in physical aggression because they are only allowed to exchange sensory communication via perforated barriers, dominant males had a lower caloric efficiency than subordinates. This is likely because dominant animals constantly signal their dominance to either females or male competitors requiring significant metabolic energy investment (*Desjardins, Maruniak & Bronson, 1973*; *Hurst & Beynon, 2004*). Using the same group housing environment we used in this study, we previously showed males with a higher social rank invest more in producing MUPs (*Lee, Khan & Curley, 2017*). Taken together, we suggest that maintaining dominant status in social hierarchies is energetically costly and animals consume more food to meet these demands. One possible common underlying mechanism linking the increased production of MUPs and feeding may be the relationship between ghrelin and growth hormone (GH). Although the regulation of food intake and energy balance is regulated by multiple neuropeptides, ghrelin directly promotes both food intake and GH release (*Gunawardane et al., 2000*). In rodents, GH directly stimulates the liver to produce MUPs (*Sagazio, Shohreh & Salvatori, 2011*; *Noaín et al., 2013*). Therefore, ghrelin may be elevated as animals perceive their social status as dominant, thus increasing food intake as well as MUP production, though this remains to be tested in future studies.

To our knowledge, this is the first time that the drinking frequency of all mice living in large social housing has been recorded with non-stop recording for a full light/dark cycle while evaluating all individuals' social status. We confirmed our hypothesis that more dominant individuals in the social hierarchy drink water more frequently, as predicted from our previous finding that individuals with higher social ranks produced a higher

volume of urine daily (*Lee, Khan & Curley, 2017*). Our finding that alpha males visit the water most frequently is also consistent with a previous study conducted using rats living in groups (*Blanchard & Blanchard, 1989*). Using the visible burrow system of housing four males and four female rats in a large arena, the alpha male in each group drank water significantly more often than the other three males. Interestingly, we show that alpha males drink most frequently even at the onset of social housing and this could suggest either that individual drinking behavior correlates with competitive ability or that mice are highly capable of recognizing current social context and quickly adapt their non-social behavior and physiology. We also found that subdominant individuals drank more frequently than subordinates. While alpha males in hierarchies increase their drinking frequency to match increased urination volume (*Lee, Khan & Curley, 2017*), non-alpha males require less water as they inhibit scent-marking behavior. Previous studies have shown that subordinate rats and mice limit their scent-mark to the edge of housing (*Desjardins, Maruniak & Bronson, 1973*; *Adams, 1976*; *Hou et al., 2016*) to avoid conflict with dominants (*Jones & Nowell, 1973*). This inhibition of urination could be more accentuated in subordinate males than subdominants, thus subdominants show higher drinking frequency than subordinates. It is also possible that subdominants may be primed to take-over alpha status and already increase their water intake in readiness (*Williamson, Romeo & Curley, 2017*; *Williamson et al., 2018*).

Another possible explanation for the finding that subordinate mice eat and drink less frequently is that they experience higher levels of social stress leading to appetite loss (*Meerlo et al., 1996*; *Becker et al., 2007*). We have previously found that subordinate mice have elevated corticosterone levels than alpha males only in groups with highly despotic males, suggesting that differences in social stress may not account for differences in feeding behavior. Further, social stress may also be related to increase rather than decrease in food intake (*Bhatnagar et al., 2006*; *Foster et al., 2006*; *Chuang et al., 2011*) suggesting that a complex relationship between stress and appetite exist in mice. Another alternative hypothesis is that subordinates avoid foraging when the alpha male is actively foraging to decrease their risk of attack. However, we found that foraging associations between non-alpha and alpha males were not different from those between two non-alpha males suggesting that individuals did not actively avoid the alpha male specifically. Consistent with this interpretation, we found that although many individuals had a location preference for foraging, location preference was unrelated to the alpha male's location preference in their hierarchy. Significantly, however, we did find that mice associate less strongly while foraging with any individuals that they had being in aggressive interactions with. Although we do not know the mechanism through which this behavioral pattern is achieved, it is possible that it occurs via individuals socially monitoring those other individuals that direct aggressive behavior towards them (*Alanärä, Burns & Metcalfe, 2001*; *Deaner, Khera & Platt, 2005*; *Pannozzo et al., 2007*). This finding also suggests that mice living in social hierarchies are socially competent being able to recognize each mouse and flexibly adjust their behavior based on specific social experiences.

We also show that mice visit food and water dispensers more frequently during the dark phase compared to the light phase of the light cycle. This is consistent with previous

findings that mice are more active and intake more food during the 'active' dark phase (*Ramsey et al., 2009*; *Melhorn et al., 2010*). Interestingly in the light phase, mice in stable hierarchies ate and drank more frequently compared to mice in the initial phases of group housing. This suggests that as groups become familiar with each other, individuals adjust the circadian patterning of foraging behavior. We investigated whether these shifts in temporal dynamics were different between ranks but found no difference in the proportion of time spent foraging in the light versus dark phases between dominant, subdominant or subordinate mice. This finding is in contrast to some other species such as fish and rats where it has been shown that subordinate individuals do temporally segregate their foraging from more dominant individuals (*Alanärä, Burns & Metcalfe, 2001*; *Melhorn et al., 2010*). For example, subordinate, but not dominant, rats in the visible burrow system have been found to increase their meal frequency during the light phase and decrease during the dark phase in established hierarchies (*Melhorn et al., 2010*). We did however identify that the longest period of inactivity between foraging bouts was significantly shorter for alpha males (mean 198 min) than for other males (mean 336 min). For the vast majority of individuals, the longest period of inactivity occurs during the light phase and is likely when individuals are engaged in sleep. These results suggest the possibility that dominant alpha males have significantly reduced sleep, though further studies are necessary to test how social status modulates the type, length and quality of sleep. Since sleep has restorative functions such as the removal of toxins from the brain and boosting the immune system (*Xie et al., 2013*; *McEwen & Karatsoreos, 2015*), shortened sleep pattern of alpha males may add a higher allostatic load to dominants on top of their increased metabolic needs.

## CONCLUSIONS

In this study we demonstrate how individual social status associates with feeding and drinking behavior in social hierarchies of male mice. In combination with our lab's previous findings showing the dramatic increase in MUP production and daily urination volume by alpha males, we propose that maintaining alpha status in social groups is metabolically expensive and requires dominant male mice to consume more food and increase water intake. This dynamically changes their temporal patterning of foraging behavior and may influence the behavioral patterns of other individuals in their social group. Additionally, we also show that outbred CD-1 mice are able to flexibly adapt their foraging behavior based on past agonistic interactions suggesting a degree of social competence. We believe these current results lay a basis for future studies examining the neurobiological and physiological mechanisms connecting perception of social status and critical physiological adaptions that occur during the establishment and maintenance of social hierarchies.

## ACKNOWLEDGEMENTS

We thank Dr. Rae Silver and Dr. Frances Champagne for advice and suggestions in writing the manuscript and Curley Lab students for help with behavioral observations.

### Funding

This work was supported by the Department of Psychology, Columbia University (James P Curley, Eilene Yang), Columbia University Dean's fellowship (Won Lee) and Samsung Scholarship Foundation (Won Lee). The funders had no role in study design, data collection and analysis, decision to publish, or preparation of the manuscript.

### Grant Disclosures

The following grant information was disclosed by the authors:
Department of Psychology, Columbia University.
Columbia University Dean's fellowship.
Samsung Scholarship Foundation.

### Competing Interests

The authors declare there are no competing interests.

### Author Contributions

- Won Lee conceived and designed the experiments, performed the experiments, analyzed the data, prepared figures and/or tables, authored or reviewed drafts of the paper, approved the final draft.
- Eilene Yang analyzed the data, approved the final draft.
- James P. Curley conceived and designed the experiments, analyzed the data, contributed reagents/materials/analysis tools, prepared figures and/or tables, authored or reviewed drafts of the paper, approved the final draft.

### Animal Ethics

The following information was supplied relating to ethical approvals (i.e., approving body and any reference numbers):

All experiments were conducted with approval from the Columbia University Institutional Animal Care and Use Committee (IACUC Protocols: AC-AAAP5405, AC-AAAG0054).

### Data Availability

All raw data and code used in this paper are publicly available at GitHub https://github.com/jalapic/foraging.

### Supplemental Information

Supplemental information for this article can be found online at http://dx.doi.org/10.7717/peerj.5617#supplemental-information.

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
