# Peer review of "Foraging dynamics are associated with social status and context in mouse social hierarchies"

_PeerJ, doi:10.7717/peerj.5617_

## Round 0.1 · original submission · Major Revisions

The reviewers have comments demanding revision. I suggest that yiu take more time for revision and explain and discuss all the necessary details of the behavior experiments.

Reviewer 1 ·

Basic reporting

No comment. Article meets all standards described.

Experimental design

The research question is relevant and meaningful, and the methods well defined, and the study was performed to a high standard.

The methods are clearly described, and the results are similarly detailed. Appropriate statistics were undertaken, and all necessary information is present in either the manuscript or raw data. The discussion/conclusions resist making too many assumptions or large claims, though I do appreciate the authors interpreting their data in a thoughtful and clever way.

Validity of the findings

The data are robust, sound, and well controlled. The conclusions were reasonable based upon the data presented. Any speculation was clearly indicated as such.

Additional comments

Lee et al. provide an interesting manuscript that describes how social status affects foraging behaviors in novel complex social housing environment, which the Curley lab has established and validated. Briefly, they demonstrate that alpha males (those at the top of the hierarchy) are the mice that eat and drink most frequently, and also showed reduced quiescence of foraging (during the light phase), even prior to the formation of stable hierarchies. One explanation the authors make for this is the increased metabolic costs of maintaining their high status in the hierarchy. Related, they demonstrate that agonistic behavior corrected for gregariousness (using HWIG) demonstrated that animals change their foraging behavior such that they will avoid conspecifics that they have engaged with in aggressive situations.
I have only very minor suggestions/comments.

In figures 1 and 3, I assume that the authors assigned status in “day 1” (i.e. BEFORE the hierarchy was established) by what the animal’s behavior was AFTER the establishment of the hierarch. That is, the alpha male was decided after the hierarchy was stable, and this mouse was then indicated as “alpha” on the “day 1” graphs. If this is the case, perhaps making it clearer would be beneficial to readers.

Related, I found the figure captions a bit too short – I’m wondering if adding more information to those captions may not also benefit readers. I was left flipping back to the results to interpret the findings.

I found one typological error (though I did not do a complete proofing) on line 403 – “more frequency” likely should be “more frequently”.

Overall, this was a very intriguing study to read, with results that are important for other researchers to understand when examining social dynamics.

Reviewer 2 ·

Basic reporting

The authors introduced a new interesting model to investigate social interactions and formation of a social hierarchy.
However, for better understanding of the article some corrections should be done.
At first, figures should be more appropriately described and labeled. The authors should explain what is indicated on the axises. For example, what does "Eating frequency" mean? The number of events? What is represented on box-plots - mean, median...? This applies to all figures.
I’m not a native English speaker so I cannot check language aspects; I can only give some advice. In Method section, line 159, the authors used the word “coders” but it is better to say “observers”.
Line 59 – check the misspelling

Experimental design

The experimental design raises several questions:
- Why only 5 cohorts were analyzed on Day1 (and 11 cohorts tested later)? The authors attempted to analyze the influence of stable and unstable hierarchies on behavioral parameters but they limited one of their groups by 5 observations.
- The groups with a stable hierarchy have different time since group formation - it introduces an additional factor in the statistical model. The reason for this is not explained. And I didn't find the information in the article about how the authors define a stable hierarchy. In 2 cohorts alpha males changed their status at the end of group housing, can we regard this hierarchy as stable? Maybe you need to exclude these cohorts from experiment?

Validity of the findings

The authors should explain the choice of the methods of statistical analysis with using a Bayesian Markov chain Monte Carlo sampling.

---

## Round 0.2 · accepted · Accept

It took time for second revision. Sorry for some delay. Now the reviewers have no more remarks.

# Reviewer 1 ·

Basic reporting

The authors have addressed all of my comments/concerns.

Experimental design

The authors have addressed all of my comments/concerns.

Validity of the findings

The authors have addressed all of my comments/concerns.

Additional comments

The authors have thoughtfully addressed all of my comments/concerns.

Reviewer 2 ·

Basic reporting

Revised article became more understandable to the readers.

Experimental design

Methods described with sufficient detail & information to replicate.

Validity of the findings

Data is robust, statistically sound, & controlled.

Additional comments

All questions arose in the previous review were explained. Revised article became more understandable to the readers.
The article meets the PeerJ criteria and should be accepted.